# Prescriber Commitment Posters to Increase Prudent Antibiotic Prescribing in English General Practice: A Cluster Randomized Controlled Trial

**DOI:** 10.3390/antibiotics9080490

**Published:** 2020-08-07

**Authors:** Anna Sallis, Paulina Bondaronek, Jet G. Sanders, Ly-Mee Yu, Victoria Harris, Ivo Vlaev, Michael Sanders, Sarah Tonkin-Crine, Tim Chadborn

**Affiliations:** 1Public Health England Behavioural Insights (PHEBI), Research, Translation & Innovation, Public Health England, Wellington House, 133-155 Waterloo Rd, Lambeth, London SE1 8UG, UK; paulina.bondaronek@phe.gov.uk (P.B.); j.g.sanders@lse.ac.uk (J.G.S.); tim.chadborn@phe.gov.uk (T.C.); 2eHealth Unit, Research Department of Primary Care and Population Health, University College London, London SE1 8UG, UK; 3Department of Psychological and Behavioural Science, London School of Economics and Political Sciences, London SE1 8UG, UK; 4Nuffield Department of Primary Care Health Sciences University of Oxford, Radcliffe Primary Care, Radcliffe Observatory Quarter, Woodstock Road, Oxford OX2 6GG, UK; ly-mee.yu@phc.ox.ac.uk (L.-M.Y.); victoria.harris@phc.ox.ac.uk (V.H.); sarah.tonkin-crine@phc.ox.ac.uk (S.T.-C.); 5Warwick Business School, University of Warwick, Coventry CV4 7AL, UK; ivo.vlaev@wbs.ac.uk; 6The Behavioural Insights Team. 4 Matthew Parker St, Westminster, London SW1H 9NP, UK; michael.sanders@nesta.org.uk; 7NIHR Health Protection Research Unit in Healthcare Associated Infections and Antimicrobial Resistance, University of Oxford, Oxford OX2 6GG, UK

**Keywords:** primary care, antimicrobial stewardship, antibiotic resistance, commitment posters, antibiotic prescribing

## Abstract

Unnecessary antibiotic prescribing contributes to Antimicrobial Resistance posing a major public health risk. Estimates suggest as many as half of antibiotics prescribed for respiratory infections may be unnecessary. We conducted a three-armed unblinded cluster randomized controlled trial (ISRCTN trial registry 83322985). Interventions were a commitment poster (CP) advocating safe antibiotic prescribing or a CP plus an antimicrobial stewardship message (AM) on telephone appointment booking lines, tested against a usual care control group. The primary outcome measure was antibiotic item dispensing rates per 1000 population adjusted for practice demographics. The outcome measures for post-hoc analysis were dispensing rates of antibiotics usually prescribed for upper respiratory tract infections and broad spectrum antibiotics. In total, 196 practice units were randomized to usual care (*n* = 60), CP (*n* = 66), and CP&AM (*n* = 70). There was no effect on the overall dispensing rates for either interventions compared to usual care (CP 5.673, 95%CI −9.768 to 21.113, *p* = 0.458; CP&AM, −12.575, 95%CI −30.726 to 5.576, *p* = 0.167). Secondary analysis, which included pooling the data into one model, showed a significant effect of the AM (−18.444, 95%CI −32.596 to −4.292, *p* = 0.012). Fewer penicillins and macrolides were prescribed in the CP&AM intervention compared to usual care (−12.996, 95% CI −34.585 to −4.913, *p* = 0.018). Commitment posters did not reduce antibiotic prescribing. An automated patient antimicrobial stewardship message showed effects and requires further testing.

## 1. Introduction

Antimicrobial resistance (AMR) is one of the major risks facing public health [1,2]. Globally, AMR is associated with approximately 700,000 deaths annually [3]. One of the major modifiable factors contributing to AMR is the unnecessary use and overuse of antibiotics [4,5,6]. In England, in 2017, 81% of all antibiotics prescribed were in primary care [7], with an estimated 46% of antibiotics prescribed for respiratory tract infections [8]. As many as half of the antibiotics prescribed for respiratory infections may be unnecessary [9,10].

Public misconceptions about the efficacy of antibiotics for treating self-limiting infections are prevalent [11,12,13]. Moreover, patient expectations, whether real or perceived, have an impact on the clinician’s prescribing behavior and can lead to unnecessary prescribing [14,15,16]. Although a recent review suggests that, globally, patient expectations for the receipt of antibiotics for respiratory tract infections are declining, it remains important for antimicrobial stewardship efforts to tackle patient expectations for antibiotics and support clinicians to make clinically appropriate prescribing decisions [17].

Interventions targeting the public mainly focus on education about the consequences of antibiotic consumption [18]. Although these interventions can be effective in increasing knowledge and understanding of AMR [19,20], evidence for their effectiveness in terms of influencing behavior is limited [21,22] and mixed [23], and may potentially produce some unintended consequences [24]. The World Health Organization’s global survey of antibiotic awareness campaigns recommends that intervention content aimed at the public adheres to scientific evidence, is tailored to the context, and is based on behavior change theory [21].

Literature has emphasized the need for multifaceted interventions targeting multiple influences on behavior to promote prudent antibiotic use [25]. Several have been trialed and some have shown reductions in antibiotic prescribing [26,27,28,29,30]. Interventions have commonly included skills training for clinicians, for example, in enhanced communication [29,30] and the use of point-of-care diagnostics, such as C-Reactive Protein [31]. However, interventions like these require a significant amount of training time for clinicians and may not be appealing or accessible to those with high workloads.

As an alternative approach requiring less effort on the part of the prescriber, clinician commitment posters displayed in Californian consulting rooms delivered a 19.7 absolute percentage point reduction in antibiotic prescribing across five primary care clinics and 954 patients [26]. The clinician commitment posters were poster-sized letters from clinicians to patients stating the clinicians’ commitment to appropriate antibiotic prescribing. Posters were signed by the clinician and included the clinician’s photograph. Public commitments aim to influence behavior by increasing the salience and stability of the intentions relevant to the behavior [32]. Commitment making has been linked to an individual’s need for self-consistency (a motivation to align our behavior with previous decisions, especially those which are made public) [33]. In addition, the motivation to protect one’s public image may result in a stronger motivation to adhere to the public commitment [34]. It should be noted that the poster is expected to ‘work’ through the clinicians’ commitment to the behavior described (in this case, appropriate antibiotic prescribing). It can also support clinicians’ decisions during consultations as it can be referred to during consultations with patients. It is not intended to be a public health messaging poster aimed at patients. The simplicity and success of this intervention creates potential for a low-cost, scalable intervention and warrants testing in other contexts.

Publicly displayed commitment posters target both clinicians and patients simultaneously, which may help to align perceptions and expectations. A recent review on the behavioral drivers of patient antibiotic consumption proposed that interventions which require less reflective thought and more environmental cues on how to respond to symptoms of self-limiting infections could have good effects on behavior [20]. One example proposed was to use appointment-booking systems in General Practices as an opportunity to prime patients about appropriate antibiotic use by informing expectations for antibiotic prescriptions prior to the consultation. For example, by playing an automated antimicrobial stewardship message to patients before they reach the stage of appointment booking. Such point-of-choice prompts involve the disruption of habitual behavior through a change in contextual cues to action. The prompt interrupts automatic thinking and may lead to the substitution of one behavior for another [35]. This ‘bottle neck’, which every patient needs to pass through in order to access a GP and antibiotics, offers another low cost, scalable intervention giving patients the opportunity to consider alternative ways to respond to their symptoms. Targeting both patients and prescribers with antimicrobial stewardship interventions might be a more effective approach to reducing antibiotic prescribing and consumption than targeting just one group.

This study aimed to test a commitment poster to reduce antibiotic prescribing in the context of English General Practice. Additionally, we tested an automated antimicrobial stewardship message played to patients on telephone appointment booking lines, aimed at adjusting patient expectations for antibiotics.

## 2. Materials and Methods

### 2.1. Study Design, Participants, and Setting

This three-armed cluster randomized controlled trial invited interest from GPs via 42 Clinical Commissioning Groups (CCGs). The trial protocol is shown in Appendix A. The inclusion criteria were that practices had an automated call answering system, practices had a maximum of two prescribers per consulting room, fewer than 20% of appointment bookings were made online, and it was not a walk-in center. CCGs were excluded if fewer than three practices agreed to participate. The trial was registered with the ISRCTN trial registry number 83322985.

### 2.2. Randomization

In December 2015, practices were randomly allocated to control, Commitment Poster (CP), or CP and automated message (CP&AM) groups, stratified by CCG as follows. Each practice, within each CCG, was randomly assigned a number between 0 and 1. The first 40% of practices occurring numerically in the list were allocated to the control, and approximately 30% were allocated to CP and CP&AM respectively thereafter. Quotas were selected based on the number of participating practices per CCG to ensure close to a numerical balance in the number of practices assigned. In total, 17 practices were combined into 2 groups of 2, 1 group of 3, and 1 group of 10 for randomization due to co-located prescribers or shared consulting rooms. Prescribers and researchers were not blinded to trial arm allocation because of the nature of this behavioral intervention.

### 2.3. Procedure and Interventions

The trial ran from 1 February 2016 to 30 July 2016. Prescribers in intervention practices were instructed to read the template CP and provide their photograph and hand-written name to researchers, who created and posted a personalized version to be displayed in consultation rooms. The poster is provided in Appendix A. Signatures were not requested due to the potential for misuse. The poster was based on the CP tested by Meeker et al. [26] and an antibiotic guardian poster (English Primary Care antimicrobial stewardship campaign aimed at healthcare professionals) [36]. Practices in the CP&AM arm were additionally asked to play an automated message on all practice telephone lines leading to appointment bookings stating the following: *“GPs in this practice do not prescribe antibiotics for infections which usually get better on their own such as colds and flu. Please visit your pharmacist for advice.”* Practices were instructed to play the message before or after existing messages during opening hours. Interventions are described using the Behaviour Change Technique Taxonomy (BCT-T) V1 in Appendix A.

### 2.4. Fidelity Checks

Intervention practices were contacted by post and asked to confirm the receipt and display of posters approximately two months after the trial started. Implementation of the automated message was checked at the start and mid-point of the trial by phoning practices.

### 2.5. Data Collection and Outcome Measures

Data were extracted from the national administrative ePACT database [37]. This covers National Health Service England prescriptions written in England which have been dispensed in the UK. Aggregate data for each month are collected at a practice level only and are not available for individual prescribers; indication for antibiotic therapy is not recorded.

The primary outcome measure was antibiotic item dispensing rates for the pooled data during the intervention period per 1000 population adjusted for sex and age (with the population determined by the practice list size). The secondary outcome measures were dispensing rates of antibiotics usually prescribed for upper respiratory tract infections (URTI) (penicillins and macrolides) and three classes of broad-spectrum antibiotics (Co-amoxiclav, Cephalosporins, and Quinolones).

### 2.6. Power Calculation

ePACT data from March 2015 was used to determine the sample size. After excluding outliers, practices had a mean dispensing rate per 1000 head of population of 79.54 and a standard deviation (SD) of 20.02. As we intended to use a fixed effects regression model accounting for time trends with seasonal effects and making use of the previous period’s data for each practice, we controlled these to estimate our minimum detectable effect size (MDES). A sample of 177 practices across three arms would provide an 80% power and 5% level of significance (two-sided) to detect an MDES of 3.09 doses per 1000 population (i.e., 3.9% reduction).

### 2.7. Statistical Analysis

The primary difference-in-difference analysis on the intent-to-treat population used a linear fixed effects regression model accounting for time trends with seasonal effects, making use of the previous period’s data to estimate the effect of the treatment status on prescribing. The primary analysis controlled for CCG, the number of GPs in the practice, the practice population size, and practice demographics (proportion within each gender/age category). Robust standard errors were used to account for the clustering of practices within CCGs. The secondary analysis fitted models pooling the two interventions into a single model comparing each with usual care. As no practices delivered the AM alone, the first effects in the pooled analysis were interpreted as the effect of receiving the CP intervention and any additional effect of receiving the AM intervention. Two per-protocol analyses were carried out, firstly by excluding randomized practices who dropped out before the start of the trial and secondly by excluding practices with a poor fidelity (i.e., no response to fidelity checks or fewer than 100% of participating GPs per practice displaying posters or not implementing the AM). Models controlled for the number of GPs in the practice, the practice population size, and practice demographics. We explored the moderation effect of treatment by practice/participant demographics using the same model as the primary analysis, with an additional main effect and an interaction. The effects on the number of doses of broad-spectrum antibiotics and number of antibiotics typically prescribed for URTI were explored in a subgroup analysis. Analyses were performed using STATA version 15.1.

## 3. Results

Across 29 CCGs, 209 practices were randomized as 196 units: Control *n* = 60; CP *n* = 66; CP&AM *n* = 70 (Figure 1). Some practices (CP *n* = 6; CP&AM *n* = 12) dropped out post-randomization, prior to the intervention’s start. The total number of GPs registered to participating practices and therefore contributing prescribing data to the main outcome measure was 1842 (CP *n* = 555, CP&AM *n* = 566, and Control *n* = 721). The total number of prescribers opting into the study (recorded for trial arms only) was 889 (CP *n* = 442, response rate 79.64%; CP&AM *n* = 447, response rate 78.98%).

### 3.1. Fidelity Checks

Figure 2 describes the responses to poster fidelity checks for the 889 participating GPs in the two trial arms. In total, 15 (CP) and 12 (CP&AM) practice units did not respond to poster fidelity checks, representing missing data for 119 and 87 prescribers, respectively. Of the practice units responding, a further 11 (CP) and 61 (CP&AM) prescribers were reportedly not displaying the posters.

All practices in the first per protocol analysis implemented the AM at the start of the trial (i.e., excluding the 12 who dropped out immediately post-randomization). Midway through, seven practices had stopped the message (three unknown cause, one technical issue, two immediate responses at reception interrupting the phone message, and one building work at the clinic), with five reinstating the message when requested. The two practices not reinstating were excluded for the second per protocol analysis.

### 3.2. Overall Effect of Interventions Compared to the Control

An average of seven practices took part per CCG (minimum of 3 and maximum of 20), with a mean baseline dispensing rate of 463 antibiotics per 1000 population and an average patient practice population size of 9501. The mean number of GPs per practice was 11.5. Adjusting for practice demographics, 5.67 more items were estimated to have been dispensed per 1000 population in the CP arm (Table 1) and 12.58 fewer items per 1000 population were predicted to have been dispensed in the CP&AM intervention compared to the control, but neither difference was statistically significant (Table 2).

Like the primary analysis, the results of the pooled analysis show a non-significant detrimental effect of the CP (5.904, 95% CI −8.861−20.669, *p* = 0.420) and a beneficial effect of the AM, with the latter reaching statistical significance (−18.44 fewer antibiotics items dispensed, 95% CI −32.60 to −4.29, *p* = 0.01) (Table 3). Secondary analyses found no significant moderation effect by practice population, number of GPs in a practice, or practice population demographics (proportion of the population by gender or age).

### 3.3. Effect of Interventions on Antibiotics Commonly Prescribed for URTI and Broad Spectrum Prescribing

There was no statistically significant effect of the intervention arms on the number of broad spectrum antibiotics dispensed for CP (2.52, 95% CI −1.73 to 6.77, *p* = 0.24) or CP&AM (−0.31, 95% CI −3.86 to 3.24, *p* = 0.86) and no evidence of an impact of CP on the number of antibiotics typically prescribed for URTIs (AB Items 1.97, 95% CI −7.59 to 11.54, *p* = 0.68). However, 13.00 fewer macrolides and penicillins per 1000 population were prescribed in the CP&AM intervention compared to the control, which was statistically significant (−13.00, 95% CI −34.59 to −4.91, *p* = 0.02).

No differences in the primary analysis were observed in the per-protocol populations.

## 4. Discussion

This study assessed the effects of a prescriber commitment poster in consulting rooms to encourage clinically appropriate antibiotic prescribing, alone and in addition to an automated message on patient appointment booking phone lines. The primary analysis showed no statistically significant differences in antibiotic prescribing between the interventions and usual care. However, a statistically significant reduction in antibiotic prescribing was observed in response to the answer phone message when secondary analysis separated the effects from the commitment poster in a single model. Furthermore, sub-group analysis showed that the combined intervention caused a statistically significant reduction in the prescribing of antibiotics commonly used for upper respiratory tract infections.

This is the first study to test an antibiotic prescribing commitment poster at scale in an English primary care setting. In contrast to the 19.7% reduction in unnecessary antibiotic prescribing seen using a similar intervention in the US, we found no such effects. Smaller effects were expected in the present study, as we measured overall antibiotic prescribing rather than “inappropriate” prescribing, as in the original study by Meeker et al. [26]. However, this difference is unlikely to explain the lack of effect observed.

The commitment poster targets a generalized behavioral insight that humans are motivated to act in line with their public commitment to achieve self-consistency as viewed by the self and others [38,39,40], as opposed to directly targeting any of the known influences on antibiotic prescribing (e.g., perception of a limited consultation time and high workload, concern about adverse patient events, fear of legal issues, lack of skills to reassure and educate patients, diagnostic uncertainty, and patient expectations) [41,42,43], although it may have supported GPs in communicating to patients reasons for not prescribing. Whilst many of these reported barriers to prudent antibiotic prescribing are similar in England and the US, one important difference is the consultation length, with a norm of 10 min in England compared to 20 min in the US. Therefore, it is possible that despite the good intentions, UK prescribers were unable to enact their commitments due to competing demands during the consultation and workload pressures, as has frequently been cited by UK GPs [44,45]. A short consultation length has previously been associated with the overuse of antibiotics [46,47,48].

Commitments are aimed at improving intentions to enact a particular behavior. However, a commitment poster alone does not facilitate the translation of intention into action and does not overcome any barriers to implementation. Both specific action planning and increasing confidence in enacting these actions are mediators increasing the likelihood of engaging in the desirable behavior [49,50,51,52]. An improved intervention could use action planning to facilitate the GPs to create a plan to overcome any barriers to implementing the commitment (e.g., a limited consultation time) and increase their self-efficacy for taking these actions.

The US sample included 14 prescribers from five clinics, whereas the English sample included almost 900 prescribers from over 200 practices. To test a cost-effective and scalable equivalent intervention, it was not feasible to exactly replicate the US trial procedure of a research coordinator being onsite at clinics to take photos and gain the consent of clinicians in person (D. Meeker, personal communication, August 11, 2017). Hence, we cannot be certain that clinicians saw the poster and actively committed to its contents or whether this was always completed by an intermediary. This lower level of engagement in the intervention may have affected the prescriber’s engagement in the commitment. Research shows that active public commitments (e.g., the act of signing the poster) are more effective than passive commitments [39,40]. This is an important consideration because versions have been available to download from the Internet [53], with the instructions to add the healthcare professional photo and signature [54]—this could be done by either clinicians or administrators.

When using an automated antimicrobial stewardship message for patients intending to book appointments, in addition to the poster, fewer prescriptions for antibiotics commonly used for URTI were filled. The automated message directly addressed patient’s expectations regarding antibiotic prescribing for colds and the flu [16,55], explaining that ‘*GPs in this practice do not prescribe antibiotics for infections which usually get better on their own such as colds and flu’* [10]. Reduced prescribing may have been due to patients seeking pharmacy advice instead of booking a consultation, although such a conclusion would require different outcome measures which allow patients to be followed up or local pharmacies. The effect of the automated message alone should be tested further given the increasing number of appointments booked online. In addition, due to the exponential increase in the use of online consultations due to Covid−19, an on-screen prompt should additionally be included in any testing.

This study involved a large national sample of prescribers with interventions delivered remotely, allowing the isolation of intervention effects (i.e., limited influence of the research team on outcomes), using routinely collected data with a minimal impact on GPs’ workload and at a low cost. The limitations of this study include an insufficient sample size for a factorial design, and the inability to interpret this study as a direct replication due to additional elements in the poster and delivery mode of the intervention. Indeed, the remote delivery of the intervention may have impacted the fidelity with which the intervention was implemented—fidelity checks were self-reports about whether the poster was displayed correctly, as opposed to whether the prescriber engaged in the commitment poster. The sample was a convenience sample and the practices may not represent national practices (for example, antibiotic prescribing rates in our sample increased in 2016, whereas nationally, the rate was declining [7]). Additionally, as it was not possible to determine treatment indication like in the Meeker et al. trial, it is possible that this contributed to the difference in findings. A process evaluation with prescribers who took part in this study is reported separately. A final limitation is the high drop-out rate post-randomization. These limitations highlight the difficulties in conducting pragmatic research involving General Practice recruitment and in delivering interventions with a high fidelity, remotely and at scale.

## 5. Conclusions

This study was a large-scale pragmatic trial conducted within the National Health Service primary care setting in England testing low-cost behavioral interventions’ ability to influence behavior. We were not able to detect an effect of a prescriber commitment poster delivered in an English primary care setting on antibiotic dispensing. However, the addition of an automated message for patients about antibiotic prescribing for colds and the flu appears promising and requires further testing. It could easily be implemented and evaluated by individual practices. Further testing could determine whether the message deterred appointment bookings or substituted them for pharmacy visits. This study highlights the importance of attempts to replicate effective interventions in different contexts and at scale to ensure both cost-effectiveness and the delivery of active intervention components. The aim is to identify low-cost and scalable interventions that reduce the morbidity and mortality from antimicrobial resistance.

## Figures and Tables

**Figure 1 antibiotics-09-00490-f001:**
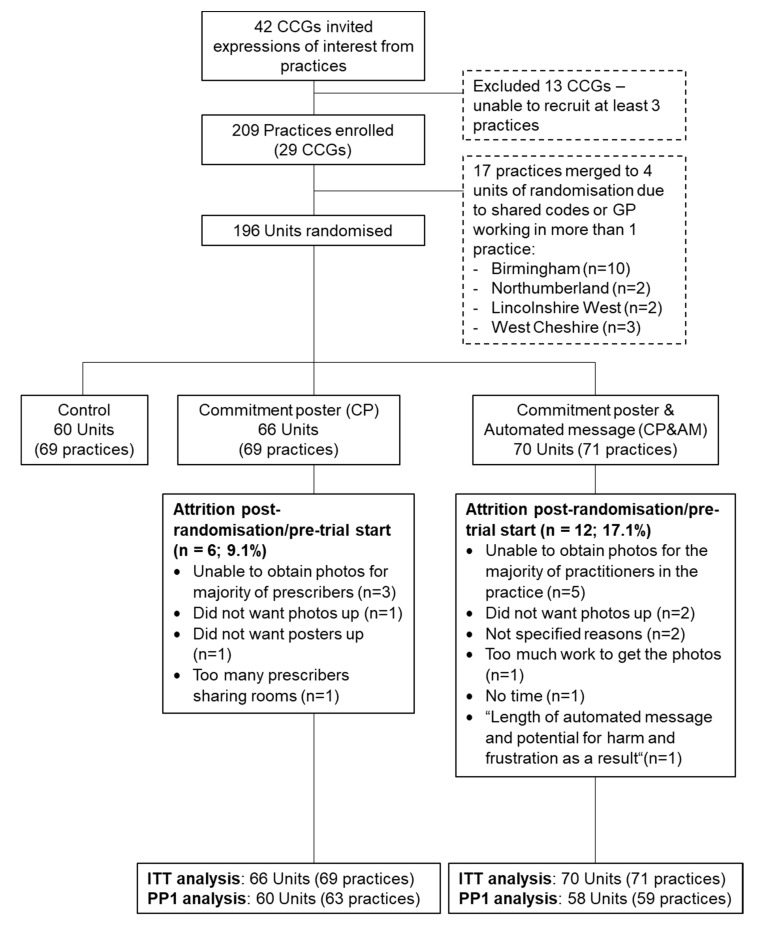
Flow of participating practices.

**Figure 2 antibiotics-09-00490-f002:**
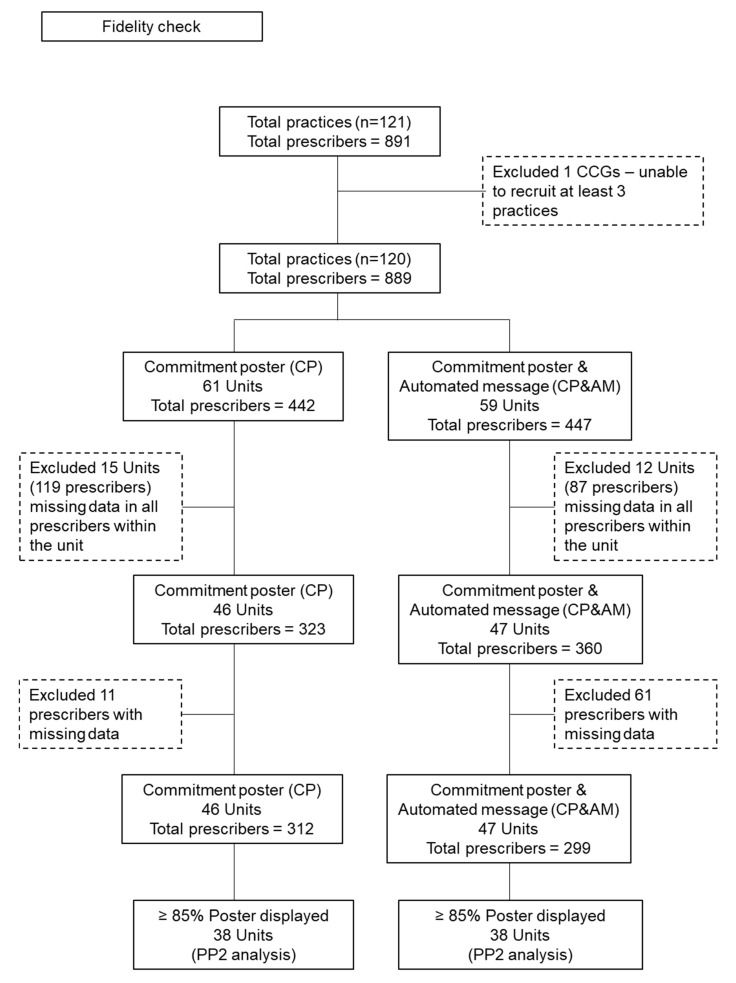
Responses to poster fidelity checks for the 889 participating GPs in the two trial arms.

**Table 1 antibiotics-09-00490-t001:** Regression estimates for the effect of the commitment poster on the number of antibiotic items prescribed.

Prescribing Rate per 1000 Population ^a^	Coefficient (β_1_)	Robust Standard Error	*p*-Value	95% C.I.
Commitment poster	5.67	7.54	0.46	−9.77 to 21.11
Baseline prescribing rate	0.78	0.03	<0.001	0.71 to 0.84
Practice population ^b^	0.01	0.00	<0.001	0.00 to 0.01
Number of GPs ^c^	−0.67	0.43	0.13	−1.55 to 0.22
February (reference)	-	-	-	-
March	32.54	4.13	<0.001	24.09 to 40.99
April	−24.16	3.06	<0.001	−30.43 to −17.88
May	−62.63	4.80	<0.001	−72.47 to −52.79
June	−59.80	4.18	<0.001	−68.36 to −51.25
July	−58.35	3.69	<0.001	−65.91 to −50.80
Year 2016 (reference 2015)	22.60	6.77	0.00	8.74 to 36.46

^a^ Adjusted for practice population demographics (proportion of the population within each gender/age category) and Clinical Commissioning Group (CCG). ^b^ Count of number of patients registered to the practice. ^c^ Count of number of GPs employed in the practice.

**Table 2 antibiotics-09-00490-t002:** Regression estimates for the effect of the commitment poster and automated message on the number of antibiotic items prescribed.

Prescribing Rate per 1000 Population	Coefficient (β_1_)	Robust Standard Error	*p*-Value	95% C.I.
Commitment poster and automated message ^a^	−12.58	8.86	0.17	−30.73 to 5.58
Baseline prescribing rate	0.75	0.05	<0.001	0.64 to 0.86
Practice population ^b^	0.01	0.00	0.01	0.00 to 0.01
Number of GPs ^c^	−0.29	0.68	0.68	−1.68 to 1.10
February (reference)	-	-	-	-
March	33.66	3.83	<0.001	25.81 to 41.50
April	−25.80	3.92	<0.001	−33.83 to −17.77
May	−67.17	4.16	<0.001	−75.68 to −58.65
June	−58.82	4.03	<0.001	−67.07 to −50.57
July	−58.69	4.25	<0.001	−67.38 to −49.99
Year 2016 (reference 2015)	23.71	7.38	0.00	8.60 to 38.83

^a^ Adjusted for practice population demographics (proportion of the population within each gender/age category) and CCG. ^b^ Count of number of patients registered to the practice. ^c^ Count of number of GPs employed in the practice.

**Table 3 antibiotics-09-00490-t003:** Regression estimates for the effect of the pooled model.

Prescribing Rate per 1000 Population ^a^	Coef. (β_1_)	Robust Standard Error	*p*-Value	95% C.I.
Commitment Poster	5.90	7.21	0.42	−8.86 to 20.67
Intervention: Automated message ^a^	−18.444	6.91	0.01	−32.60 to −4.29
Baseline prescribing rate	0.77	0.03	<0.001	0.70 to 0.82
Practice population ^b^	0.01	0.00	0.00	0.00 to 0.01
Number of GPs ^c^	−0.13	0.35	0.72	−0.85 to 0.59
February (reference)	-	-	-	-
March	31.87	3.07	<0.001	25.59 to 38.15
April	−26.74	3.11	<0.001	−33.10 to −20.38
May	−65.71	4.00	<0.001	−73.90 to −57.52
June	−60.30	3.55	<0.001	−67.58 to −53.02
July	−60.14	3.49	<0.001	−67.30 to −52.99
Year 2016 (reference 2015)	22.24	7.12	0.00	7.64 to 36.82

^a^ Adjusted for practice population demographics (proportion of the population within each age/gender category) and CCG. ^b^ Count of number of patients registered to the practice. ^c^ Count of number of GPs employed in the practice.

## Data Availability

The datasets used and/or analysed during the current study are available from the corresponding author on reasonable request.

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
