# Peer review of "Prescriber Commitment Posters to Increase Prudent Antibiotic Prescribing in English General Practice: A Cluster Randomized Controlled Trial"

_antibiotics, 2020, doi:10.3390/antibiotics9080490_

Round 1
Reviewer 1 Report
Properly planned and implemented research on the important topic of excessive, sometimes unreasonable prescription of antibiotics. However, I have a few comments.
In terms of editing, no consistency in the use of spaces, e.g. line 52, 54, 56, 94, 224, 225, 229, in Figure 1 and 2, 274, 284, 285, 287, 289 - mainly no spaces between the letter, = and number. In addition, the signature under Figure 2. should not be in bold.
I am asking the authors to improve the literature list to be in line with Antibiotics recommendations, including the year of publication should be in bold, etc.
My main question is why the authors did not add any of the Supplementary materials listed. Please send them.
Author Response
Thank you for your review.
In terms of editing, no consistency in the use of spaces, e.g. line 52, 54, 56, 94, 224, 225, 229, in Figure 1 and 2, 274, 284, 285, 287, 289 - mainly no spaces between the letter, = and number. In addition, the signature under Figure 2. should not be in bold.
Thank you for the Reviewer’s attention to detail. We have now edited the manuscript to make it consistent throughout, for example: CP n= 6; CP&AM n= 12.The non-use of space in the CP&AM is intentional to indicate the arm of the trial. The signature under Figure 2 has also been edited.
I am asking the authors to improve the literature list to be in line with Antibiotics recommendations, including the year of publication should be in bold, etc.
The style has been now changed using the MDPI style provided on the website.
My main question is why the authors did not add any of the Supplementary materials listed. Please send them.
I am not sure why they did not attached previously. I will try to add these again to the system but otherwise I will email them to the editor.
Reviewer 2 Report
In general the manuscript is of interest, well-written, and the data and statistical analysis appears appropriate, barring a few caveats that need to be addressed in the discussion.
Specific comments:
Introduction:
- Ln 101-103: Commitment posters is not immediately understood, perhaps explain what it is first before the data showing some effectiveness. Also, for the uninitiated reader, how is a commitment poster different from general posters with public health messaging?
- Ln 115 - 123: It is not clear how this relates to public commitment posters. It seems to be a different strategy, and if it is, was it tested? Conversely the AM was not explained.
- Ln 126-128: Stated this way, the AM seems like an afterthought. I suggest including a bit of literature on this and justification for why it was included as well.
Materials & Methods
- Ln 170 - 171: The fidelity check for commitment posters is entirely by self-report?
- Ln 179: Given that the work by Meeker et al., 2016 provided information regarding indication for treatment and this work is meant to replicate to some extent the findings in the NHS setting; the fact that indication for treatment is not recorded for this study deserves more discussion.
Results
- Ln 223: Was the timeframe of study mentioned earlier in the Methods? It should. And would the time frame (6 months) be a possible reason why effects were not observed? How long does is behavior change from commitment posters expected to kick in?
Discussion
- Ln 306-310: this is the question that remains, whether the appropriateness of the antibiotic prescription could be addressed. The lack of record on treatment indication needs to be discussed further, because it raises the question whether by some coincidence the different treatment arms were possibly facing different rates of bacterial/non-bacterial infections.
- Ln 322 - 323: If this was a possible factor in prescription rates, it would have been interesting to know the average consultation length of the intervention and control arms
- Ln 343 - 345: This links back to the idea of self-report in fidelity checks and whether the CPs were actually implemented as intended.
- Ln 358-361: Given the statistical significance, it would be good to expand this section to explore further what could be possible reasoning and the different outcome measures to appropriately address the effect of AM.
Conclusions
1. Perhaps add the caveat that the “active commitment” on GPs could not be confirmed in this design

Author Response
Thank you for your review.
Introduction:
- Ln 101-103: Commitment posters is not immediately understood, perhaps explain what it is first before the data showing some effectiveness. We have added more detail about the commitment poster in the effective research trial described. See lines 101-104. Also, for the uninitiated reader, how is a commitment poster different from general posters with public health messaging? We have added a line about how they differ. See lines 110-116.
- Ln 115 - 123: It is not clear how this relates to public commitment posters. It seems to be a different strategy, and if it is, was it tested? Yes, it is a different strategy but retains the ‘nudge’ element in that it also involved an environmental cue. Lines 133-136 have been added to explain why the two strategies may work together. Conversely the AM was not explained. Lines 126-127 now provide further detail.
- Ln 126-128: Stated this way, the AM seems like an afterthought. I suggest including a bit of literature on this and justification for why it was included as well. Lines 133-136 have been added to explain why the two strategies may work together. There is no further literature on this – the idea was the result of a ‘behavioural analysis’ (a partly theoretical piece of work) by some of the same authors (Ref 20).
Materials & Methods
- Ln 170 - 171: The fidelity check for commitment posters is entirely by self-report? Yes for the posters. The process is described in lines 183-187. It would not have been practical or scalable to attend practices to check if posters were displayed. For the answer phone message we were able to check this ourselves by calling the practices.
- Ln 179: Given that the work by Meeker et al., 2016 provided information regarding indication for treatment and this work is meant to replicate to some extent the findings in the NHS setting; the fact that indication for treatment is not recorded for this study deserves more discussion. Added to lines 387-390 in limitations in discussion.
Results
- Ln 223: Was the timeframe of study mentioned earlier in the Methods? It should. The trial dates have been moved from results to procedure and interventions section – see line 167. And would the time frame (6 months) be a possible reason why effects were not observed? How long does is behavior change from commitment posters expected to kick in? In the most similar study to date (Meeker, 2016) they implemented the poster for 12 weeks. Ours was implemented for 6 months. We would have expected a greater effect over a longer period but maybe it had the reverse effect.
Discussion
- Ln 306-310: this is the question that remains, whether the appropriateness of the antibiotic prescription could be addressed. The lack of record on treatment indication needs to be discussed further, because it raises the question whether by some coincidence the different treatment arms were possibly facing different rates of bacterial/non-bacterial infections. We believe the randomisation accounts for any random differences between groups. The lack of treatment indication is now added as a limitation to the discussion (lines 387-390).
- Ln 322 - 323: If this was a possible factor in prescription rates, it would have been interesting to know the average consultation length of the intervention and control arms. This would be interesting (bot not possible as we did not collect data on this) but we would expect them to be similar given the randomisation.
- Ln 343 - 345: This links back to the idea of self-report in fidelity checks and whether the CPs were actually implemented as intended. This is now elaborated upon in lines 387-390.
- Ln 358-361: Given the statistical significance, it would be good to expand this section to explore further what could be possible reasoning and the different outcome measures to appropriately address the effect of AM. Lines 375-376 now mention potential outcome measures. The reasoning has not been speculated further as the primary thinking is given – a change in patient expectations leading to a change in patient behaviour.
Conclusions
- Perhaps add the caveat that the “active commitment” on GPs could not be confirmed in this design – Lines 400-401 do mention that further testing is required to ensure the delivery of active intervention components (i.e. the commitment aspect of the poster intervention). Note line 387 mentions that a process evaluation is reported elsewhere.
Reviewer 3 Report
I have read with a lot of interest this cluster randomized trial of ambulatory antimicrobial stewardship interventions in England. The study is well designed and the manuscript is extremely well written. I have few minor comments and suggestions for improvement.
- Introduction: Relatively long, but very effective. I felt this was necessary to introduce the topic for the average reader who may not be familiar with ambulatory antimicrobial stewardship interventions.
- Methods: Line 143: Please verify that “number between 0 and 1” is correct.
- Line 187: Cephalosporins is misspelled.
- Line 215: How was practice population size obtained? Is this available online? What about patients who did not show up to clinic for over one year, did they count?
- Results: Line 223: Verify trial dates. The tables mention 2015 as a referent. Was 2015 included in trial? If not, why did the authors feel it was necessary to adjust for the prior year?
- Tables 1-3: Please explain units for continuous variables such as practice population, number of GPs, etc.
- Tables 1-3: 2 decimal points are enough. Too many decimals make the tables busier that they need to be.
- Tables 1-3: I suggest using July as a referent rather than February. This will show the seasonal variation in antibiotic prescribing better.
- Discussion: Excellent comparisons with prior studies and potential explanations of impact of interventions. However, there should be a paragraph about limitations: non-blinding, high dropout rates, fidelity issues, etc.
- Discussion: The authors should speculate on higher antibiotic use in 2016 than 2015. Was the flu season worse in 2016 than 2015? Otherwise, this is hard to understand since ambulatory antibiotic use has been going down in the USA and most of Europe.
- Discussion: It would also be useful to dedicate a paragraph to discuss difficulties in conducting this kind of research: recruitment, retention, non-compliance, fidelity, etc.
Author Response
Thank you for your helpful comments.
- Introduction: Relatively long, but very effective. I felt this was necessary to introduce the topic for the average reader who may not be familiar with ambulatory antimicrobial stewardship interventions.
- Methods: Line 143: Please verify that “number between 0 and 1” is correct. Yes, this is correct, Stata randomly assigns each practice a number between 0 and 1 (for example, 0.0123), drawn from a uniform distribution.
- Line 187: Cephalosporins is misspelled. Thank you this is now corrected.
- Line 215: How was practice population size obtained? Is this available online? What about patients who did not show up to clinic for over one year, did they count? The practice population information is available online, and is obtained based on the number and characteristics of patients who are registeredat the practice. They remain registered until they register somewhere else, whether they attend or not.
- Tables 1-3: Please explain units for continuous variables such as practice population, number of GPs, etc. For both number of GPs and practice population the counts were used. A footnote has been added to each table. So, the coefficient in table 1 for example represents an increase in prescribing rate per 1000 of 006 for every 1 increase in practice population and a decrease of 0.668 for every one extra GP. Baseline prescribing rate is per 1000, so every 1 unit increase per 1000 at baseline is predicted to increase the prescribing rate by 0.776 units per 1000.
- Tables 1-3: 2 decimal points are enough. Too many decimals make the tables busier that they need to be. The changes suggested have now been applied to Tables 1-3.
- Discussion: Excellent comparisons with prior studies and potential explanations of impact of interventions. However, there should be a paragraph about limitations: non-blinding, high dropout rates, fidelity issues, etc. These are now mentioned in limitations at line 387-394. Blinding is not mentioned as that the nature of the intervention means that would not be feasible anyway.
- Discussion: The authors should speculate on higher antibiotic use in 2016 than 2015. Was the flu season worse in 2016 than 2015? Otherwise, this is hard to understand since ambulatory antibiotic use has been going down in the USA and most of Europe. It appears that this could be a sub-set of practices with increasing prescribing rates as you are right national trends have been going down. We speculate that the practices, due to being a convenience sample may have had increasing prescribing rates hence they signed up to the trial. This has been added to the limitations at lines 399-402.
- Discussion: It would also be useful to dedicate a paragraph to discuss difficulties in conducting this kind of research: recruitment, retention, non-compliance, fidelity, etc. Added to lines 394-397. Fidelity is not discussed too much as this is the focus of the process evaluation paper. The present paper actually suggests posters were displayed as requested. It is the process evaluation paper where we learn that potentially the commitment aspect of the poster was not well implemented. This is alluded to now at line 389.